# Hyperbolic Diffusion Recommender Model

## ABSTRACT

Diffusion models (DMs) have emerged as the new state-of-the-art family of deep generative models. To gain deeper insights into the limitations of diffusion models in recommender systems, we investigate the fundamental structural disparities between images and items. Consequently, items often exhibit distinct anisotropic and directional structures that are less prevalent in images. However, the traditional forward diffusion process continuously adds isotropic Gaussian noise, causing anisotropic signals to degrade into noise, which impairs the semantically meaningful representations in recommender systems.

Inspired by the advancements in hyperbolic spaces, we propose a novel *Hyperbolic Diffusion Recommender Model* (named HDRM). Unlike existing directional diffusion methods based on Euclidean space, the intrinsic non-Euclidean structure of hyperbolic space makes it particularly well-adapted for handling anisotropic diffusion processes. In particular, we begin by constructing a geometrically latent space grounded in hyperbolic geometry, incorporating interpretability measures to define the latent anisotropic diffusion processes. Subsequently, we propose a novel hyperbolic latent diffusion process specifically tailored for users and items. Drawing upon the natural geometric attributes of hyperbolic spaces, we restrict both radial and angular components to facilitate directional diffusion propagation, thereby ensuring the preservation of the original topological structure in user-item interaction graphs. Extensive experiments on three benchmark datasets demonstrate the effectiveness of HDRM. Our code is available at https://anonymous.4open.science/status/HDRM-ECFA.

## CCS CONCEPTS

• **Information systems → Recommender systems**.

## KEYWORDS

Diffusion Model, Hyperbolic Spaces, Geometric Constraints

**ACM Reference Format:**
Anonymous Author(s). 2018. Hyperbolic Diffusion Recommender Model. In *Proceedings of Make sure to enter the correct conference title from your rights confirmation emai (Conference acronym 'XX)*. ACM, New York, NY, USA, 16 pages. https://doi.org/XXXXXXX.XXXXXXX

## 1 INTRODUCTION

Diffusion models (DMs) [14, 38–40] have emerged as the new state-of-the-art family of deep generative models. They have broken the

long-time dominance of generative adversarial networks (GANs) [10] in the challenging task of image synthesis [6, 14, 40] and have demonstrated promise in computer vision, ranging from video generation [13, 15], semantic segmentation [2, 44], point cloud completion [28, 67] and anomaly detection [55, 64].

Despite the increasing research on diffusion models in computer vision [6, 14, 26, 36, 40, 45], their potential in recommender systems has not been equally explored. Generative recommender models [24, 48, 50, 54, 60, 66] aim to align with the user-item interaction generation processes observed in real-world environments. Unlike other earlier generative recommender models like VAEs [24, 54] and GANs [48, 60], diffusion recommender models [21, 50, 66] leverage a denoising framework to effectively reverse a multi-step noising process to generate synthetic data that matches closely with the distribution of the training data. This highlights the exceptional ability of diffusion models to capture multi-scale feature representations and generate high-quality samples, while also ensuring improved stability during training. However, the aforementioned diffusion recommender models are still directly based on extensions of computer vision methods, neglecting the latent structural differences between images and items.

To gain deeper insights into the limitations of traditional diffusion models in recommender systems, we begin by investigating the fundamental structural disparities between images and items. Specifically, we apply singular value decomposition [58] to both image and graph data, and plot the resulting projections on a two-dimensional plane. Figure 1a reveals that the projected data from ML-1M exhibits strong anisotropic structures across multiple directions, whereas the projected images from F-MNIST (as seen in Figure 1b) form a relatively more isotropic distribution centered around the origin. As a result, items often exhibit distinct anisotropic and directional structures that are less prevalent in images [59]. Unfortunately, the traditional forward diffusion process continuously adds isotropic Gaussian noise, causing anisotropic signals to degrade into noise [58], which impairs the semantically meaningful representations in recommender systems.

Hyperbolic spaces are extensively regarded as the optimal continuous manifold for modeling discrete tree-like or hierarchical structures [1, 20, 37, 42], and have been widely studied and applied to various recommender tasks [5, 41, 43, 46, 57, 61, 62]. In hyperbolic spaces, the expansion of space is not uniform (i.e., isotropic), but rather depends on the position and direction. This leads to variations in the rate of change in distances between points along different directions. As shown in Figure 1c, hyperbolic spaces are well-suited to preserving the anisotropy of data due to its inherent geometric properties. Additionally, due to the infinite volume of hyperbolic space [33, 37], modeling uniformly distributed data tends to push the data features toward the boundary, thereby weakening the isotropy of the data to some extent (as seen in Figure 1d).

Inspired by the advancements in hyperbolic spaces, we propose a novel *Hyperbolic Diffusion Recommender Model* named HDRM. Unlike existing directional diffusion methods based on Euclidean

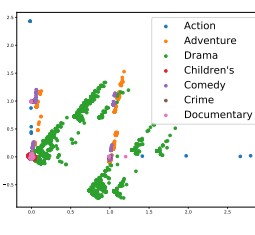

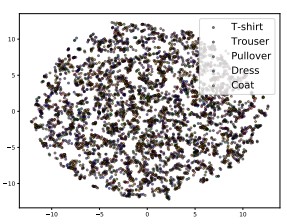

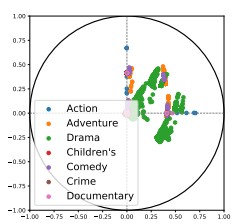

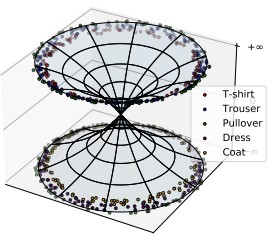

(a) ML-1M (Euclidean space)  (b) F-MNIST (Euclidean space)  (c) ML-1M (Poincaré disk)  (d) F-MNIST (Lorentz manifold)

**Figure 1: 2D visualization of the data using SVD decomposition, where each color corresponds to a unique category. (a) Euclidean visualization of the item features in MovieLens-1M; (b) Euclidean visualization of the image features in Fashion-MNIST; (c) Hyperbolic visualization of the item features in MovieLens-1M; (d) Hyperbolic visualization of the image features in Fashion-MNIST.**

space [58, 59], the intrinsic non-Euclidean structure of hyperbolic space makes it particularly well-adapted for handling anisotropic diffusion processes. In particular, we begin by constructing a geometrically latent space grounded in hyperbolic geometry, incorporating interpretability measures to define the latent anisotropic diffusion processes. Subsequently, we propose a novel hyperbolic latent diffusion process specifically tailored for users and items. Drawing upon the natural geometric attributes of hyperbolic spaces, we restrict both radial and angular components to facilitate directional diffusion propagation, thereby ensuring the preservation of the original topological structure in user-item interaction graphs. Extensive experiments on three benchmark datasets demonstrate the effectiveness of HDRM. To summarize, we highlight the key contributions of this paper as follows:

- We contribute to the exploration of anisotropic structures in recommender systems. To the best of our knowledge, this is the first work to design a hyperbolic diffusion model for recommender systems.
- We propose a novel hyperbolic latent diffusion process specifically tailored for users and items. Drawing upon the natural geometric attributes of hyperbolic spaces, we restrict both radial and angular components to facilitate directional diffusion propagation.
- Extensive experimental results on three benchmark datasets demonstrate that HDRM outperforms various baselines. Further ablation studies verify the importance of each module.

## 2 PRELIMINARIES

### 2.1 Hyperbolic Spaces

Here we introduce some fundamental concepts of hyperbolic spaces. For more detailed operations on hyperbolic spaces, please refer to Appendix A.1.

- **Manifold**: Consider a manifold $\mathcal{M}$ with $n$ dimensions as a space where the local neighborhood of a point can be closely approximated by Euclidean spaces $\mathbb{R}^n$. For instance, the Earth can be represented by a spherical space, its immediate vicinity can be approximated by $\mathbb{R}^2$.

- **Tangent space**: For every point $x \in \mathcal{M}$, the tangent space $\mathcal{T}_x\mathcal{M}$ of $\mathcal{M}$ at $x$ is set as a $n$-dimensional space measuring $\mathcal{M}$ around x at a first order.
- **Geodesics distance**: This denotes the generalization of a straight line to curved spaces, representing the shortest distance between two points within the context of the manifold.
- **Exponential map**: The exponential map carries a vector $v \in \mathcal{T}_x\mathcal{M}$ of a point $x \in \mathcal{M}$ to the manifold $\mathcal{M}$, i.e., $\exp_x : \mathcal{T}_x\mathcal{M} \to \mathcal{M}$ by simulating a fixed distance along the geodesic defined as $\gamma(0) = x$ with direction $\gamma'(0) = v$. Each manifold corresponds to its unique way of constructing exponential maps.
- **Logarithmic map**: Serving as the counterpart to the exponential map, the logarithmic map takes a point $z$ from the manifold $\mathcal{M}$ and maps it back to the tangent space $\mathcal{T}_x\mathcal{M}$, i.e., $\log_x^\kappa : \mathcal{M} \to \mathcal{T}_x\mathcal{M}$. Like $\exp_x^\kappa$, each manifold has its formula that defines $\log_x^\kappa$.

### 2.2 Diffusion Models

DMs have attained remarkable success across numerous domains, primarily through the use of forward and reverse processes [36, 50].

- **Forward Process**: Given an input data sample $x_0 \sim q(x_0)$, the forward process constructs the latent variables $x_{1:T}$ by gradually adding Gaussian noise in $T$ steps. Specifically, DMs define the forward transition $x_{t-1} \to x_t$ as:

$$
\begin{aligned}
q(x_t|x_{t-1}) &= \mathcal{N}(x_t; \sqrt{1 - \beta_t}x_{t-1}, \beta_t\mathrm{I}), \\
&= \sqrt{1 - \beta_t}x_{t-1} + \sqrt{\beta_t}\epsilon, \quad \epsilon \sim \mathcal{N}(0, \mathrm{I})
\end{aligned}
\tag{1}
$$

where $t \in \{1, \ldots, T\}$ represents the diffusion step, $\mathcal{N}(0, \mathrm{I})$ denotes the Gaussian distribution, and $\beta_t \in (0, 1)$ controls the amount of noise added at each step. This method shows the flexibility of the direct sampling of $x_t$ conditioned on the input $x_{t-1}$ at an arbitrary diffusion step $t$ from a random Gaussian noise $\epsilon$.
- **Reverse Process**: DMs learn to remove the noise from $x_t$ to recover $x_{t-1}$ in the reverse process, aiming to capture subtle changes in the generative process. Formally, taking $x_T$ as the initial state, DMs learn the denoising process $x_t \to x_{t-1}$ iteratively as follows:

$$
p_\theta(x_{t-1}|x_t) = \mathcal{N}(x_{t-1}; \mu_\theta(x_t, t), \Sigma_\theta(x_t, t)),
\tag{2}
$$

where $\mu_\theta(x_t, t)$ and $\Sigma_\theta(x_t, t)$ are the mean and covariance of the Gaussian distribution predicted by parameters $\theta$.

- **Optimization**: For training the diffusion models, the key focus is obtaining reliable values for $\mu_\theta(x_t, t)$ and $\Sigma_\theta(x_t, t)$ to guide the reverse process towards accurate denoising. To achieve this, it is important to optimize the variational lower bound of the negative log-likelihood of the model's predictive denoising distribution $p_\theta(x_0)$:

$$\mathcal{L} = \mathbb{E}_{q(x_0)}[-\log p_\theta(x_0)]$$
$$\leq \mathbb{E}_q[L_T + L_{T-1} + \cdots + L_0], \quad \text{where} \tag{3}$$

$$L_T = D_{KL}(q(x_T|x_0) \| p_\theta(x_T)),$$
$$L_t = D_{KL}(q(x_t|x_{t+1}, x_0) \| p_\theta(x_t|x_{t+1})), \tag{4}$$
$$L_0 = -\log p_\theta(x_0|x_1),$$

where $t \in \{1, 2, \ldots, T-1\}$. While $L_T$ can be disregarded during training due to the absence of learnable parameters in the forward process, $L_0$ represents the negative log probability of the original data sample $x_0$ given the first-step noisy data $x_1$, and $L_t$ aims to align the distribution $p_\theta(x_t|x_{t+1})$ with the tractable posterior distribution $q(x_t|x_{t+1}, x_0)$ in the reverse process [27].

- **Inference**: After training the model parameters $\theta$, DMs can sample $x_T$ from a standard Gaussian distribution $\mathcal{N}(0, I)$, and subsequently utilize $p_\theta(x_{t-1}|x_t)$ to iteratively reconstruct the data, following the reverse process $x_T \rightarrow x_{T-1} \rightarrow \cdots \rightarrow x_0$. In addition, previous works [22, 36] have explored the incorporation of specific conditions to enable controlled generation.

## 3 METHOD

In light of the successful applications of diffusion models [21, 50, 66], we employ a two-stage training strategy for our implementation. First, we train the hyperbolic encoder to generate pre-trained user and item embeddings. Subsequently, we proceed with the training of the hyperbolic latent diffusion process. The overall architecture is illustrated in Figure 2.

### 3.1 Hyperbolic Geometric Autoencoding

*3.1.1 Hyperbolic Graph Convolutional Network.* We adopt the hyperbolic graph convolutional network [3] as the hyperbolic encoder to embed the user-item interaction graph $\mathcal{G}_u = (\mathcal{U}, \mathcal{I})$ into a low-dimensional hyperbolic geometric space, thereby enhancing the subsequent graph latent diffusion process. The objective of the hyperbolic encoder is to generate hyperbolic embeddings for users and items. Formally, we use $\mathbf{x} \in \mathbb{R}^n$ to represent the Euclidean state of users and items. Then the initial hyperbolic state $\mathbf{e}_i^{(0)}$ and $\mathbf{e}_u^{(0)}$ can be obtained by:

$$\mathbf{e}_i^{(0)} = \exp_{\mathbf{o}}^{\kappa}(\mathbf{z}_i^{(0)}), \quad \mathbf{e}_u^{(0)} = \exp_{\mathbf{o}}^{\kappa}(\mathbf{z}_u^{(0)}), \tag{5}$$

$$\mathbf{z}_i^{(0)} = (0, \mathbf{x}_i), \quad \mathbf{z}_u^{(0)} = (0, \mathbf{x}_u), \tag{6}$$

where $\mathbf{x}$ is taken from multivariate Gaussian distribution. $\mathbf{z}^{(0)} = (0, \mathbf{x})$ denotes the operation of inserting the value 0 into the zeroth coordinate of $\mathbf{x}$ so that $\mathbf{z}^{(0)}$ can always live in the tangent space of origin.

Next, the hyperbolic neighbor aggregation is computed by aggregating the representations of neighboring users and items. Given the neighbors $\mathcal{N}_i$ and $\mathcal{N}_u$ of $i$ and $u$, respectively, the embedding

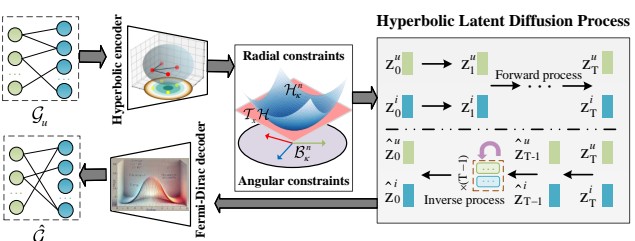

**Figure 2: An overview illustration of the HDRM architecture.**

of user $u$ and $i$ is updated using the tangent state $\mathbf{z}$ and the $k$-th ($k > 0$) aggregation is given by:

$$\mathbf{z}_i^{(k)} = \mathbf{z}_i^{(k-1)} + \sum_{u \in \mathcal{N}_i} \frac{1}{|\mathcal{N}_i|} \mathbf{z}_u^{(k-1)},$$
$$\mathbf{z}_u^{(k)} = \mathbf{z}_u^{(k-1)} + \sum_{i \in \mathcal{N}_u} \frac{1}{|\mathcal{N}_u|} \mathbf{z}_i^{(k-1)}, \tag{7}$$

where $|\mathcal{N}_u|$ and $|\mathcal{N}_i|$ are the number of one-hop neighbors of $u$ and $i$, respectively. For high-order aggregation, sum-pooling is applied in these $k$ tangential states:

$$\mathbf{z}_i = \sum_k \mathbf{z}_i^{(k)}, \quad \mathbf{z}_u = \sum_k \mathbf{z}_u^{(k)}.$$
$$\mathbf{e}_i = \exp_{\mathbf{o}}^{\kappa}(\mathbf{z}_i), \quad \mathbf{e}_u = \exp_{\mathbf{o}}^{\kappa}(\mathbf{z}_u). \tag{8}$$

Note that $\mathbf{z}$ is on the tangent space of origin. For the hyperbolic state, it is projected back to the hyperbolic spaces with the exponential map.

*3.1.2 Hyperbolic Decoder.* In accordance with these hyperbolic learning models [3, 8, 20, 33], we use the Fermi-Dirac decoder, a generalization of sigmoid, to estimate the probability of the user clicking on the item:

$$\mathbf{s}(u, i) = \frac{1}{\exp(d_{\mathcal{L}}^{\kappa}(\hat{\mathbf{e}}_0^u, \hat{\mathbf{e}}_0^i)^2 - r)/t + 1}, \tag{9}$$

where $d_{\mathcal{L}}^{\kappa}(\cdot, \cdot)$ is the hyperbolic distance as mentioned in Table 5, $\kappa$ denotes the curvature, $\hat{\mathbf{e}}_0^u$ and $\hat{\mathbf{e}}_0^i$ denote the exponential maps of $\hat{\mathbf{z}}_0^u$ and $\hat{\mathbf{z}}_0^i$ resulting from the reverse process. $r$ and $t$ are hyperparameters. Here, we slightly abuse the notation for exp: the unindexed exp refers to the exponential operation, and $\exp_{\mathbf{o}}^{\kappa}$ denotes the mapping of embeddings from the tangent space to hyperbolic space.

In summary, the workflow of hyperbolic geometric autoencoding is that the output from the encoder's final layer is projected into hyperbolic space through exponential mapping, after which the sampled latent vector is returned to Euclidean space via logarithmic mapping before being passed into the decoder layers.

### 3.2 Hyperbolic Latent Diffusion Process

Different from the linear addition in Euclidean space, hyperbolic spaces employ Möbius addition, posing challenges for diffusion on hyperbolic manifolds. Moreover, the isotropic noise causes a rapid decrease in the signal-to-noise ratio [58], making it challenging to maintain the integrity of topological structures. To achieve this,

we propose a novel hyperbolic latent diffusion process designed to tackle the aforementioned challenges.

*3.2.1 Hyperbolic Directional Diffusion.* To conserve computational resources and memory usage, we follow previous works [21, 50, 66] by clustering items during the pre-processing stage. Due to the hyperbolic nature of the space, clustering items based on their similarity to determine the diffusion direction (i.e., angle) is equivalent to dividing the hyperbolic space into multiple sectors. Next, we project the items of each cluster onto the tangent plane corresponding to its centroid, enabling the diffusion process to proceed. Formally, let $\mathbf{e}_i$ belong to the $k$-th cluster and its clustering center coordinates are $\mu_k$, then the node $\mathbf{e}_i$ is represented in the tangent space of $\mu_k$ as $\mathbf{z}_i$:

$$\mathbf{z}_{\mu_k}^i = \log_{\mu_k}^{\kappa}(\mathbf{e}_i), \tag{10}$$

where $\mu_k$ is the central point of cluster $k$ obtained by hyperbolic-kmeans [11]. Furthermore, the hyperbolic clustering parameter $k$ possesses the following characteristic:

**THEOREM 1.** *Given the hyperbolic clustering parameter $k \in [1, n]$, which represents the number of sectors dividing the hyperbolic space (disk). The directional hyperbolic diffusion is equivalent to directional diffusion in the Klein model $\mathcal{K}_{\kappa}^n$ with multi-curvature $\kappa_i \in \{|k|\}$, which is an approximate projecting onto the tangent plane set $\mathcal{T}_{\mathbf{o}_i \in \{|k|\}}$ of the centroids $\mathbf{o}_i \in \{|k|\}$.*

The proof can be found in Appendix A.2. This result elegantly illustrates the connection between our approximation algorithm and the Klein model with varying curvatures. Depending on the value of $k$, our algorithm demonstrates distinct behaviors, offering a more flexible and refined representation of anisotropy grounded in hyperbolic geometry. As a result, this improves both accuracy and efficiency in the following noise injection and training stages.

Additionally, it is important to note that a particularly fascinating feature of HDRM is the generalization of the normal distribution to Riemannian manifolds. Current approaches generally fall into two categories: the Poincaré normal distribution [31] and the hyperbolic wrapped normal distribution [32]. In the Appendix A.3, we demonstrate the non-additivity of the package normal distribution, which ultimately leads us to choose the Poincaré normal distribution.

*3.2.2 Forward Process of Geometric Constraints.* Hyperbolic spaces provide a natural and geometric framework for modeling the connection patterns of nodes during the process of graph growth [3]. Our goal is to develop a diffusion model that incorporates hyperbolic radial growth, aligning this growth with the inherent properties of hyperbolic spaces.

To ensure the maintenance of this hyperbolic growth behavior in the tangent space, we employ the following formulas:

$$q(\mathbf{z}_t^u|\mathbf{z}_{t-1}^u) = \sqrt{1-\beta_t}\mathbf{z}_{t-1}^u + \sqrt{\beta_t}\epsilon_{\mathcal{B}} + \delta\tanh(\sqrt{\kappa}\lambda_{\mathbf{z}_{t-1}^u}^{\kappa}/r)\mathbf{z}_{t-1}^u,$$
$$q(\mathbf{z}_t^i|\mathbf{z}_{t-1}^i) = \sqrt{1-\beta_t}\mathbf{z}_{t-1}^i + \sqrt{\beta_t}\epsilon_{\mathcal{B}} + \delta\tanh(\sqrt{\kappa}\lambda_{\mathbf{z}_{t-1}^i}^{\kappa}/r)\mathbf{z}_{t-1}^i, \tag{11}$$

where $\delta$ is the radial popularity coefficient that determines the diffusion strength in hyperbolic space, r is a hyper-parameter to control the speed of radial growth rate, $\epsilon_{\mathcal{B}}$ follows the Poincaré normal distribution (i.e., $\epsilon_{\mathcal{B}} \sim \mathcal{N}_{\mathcal{B}}(0, \mathbf{I})$), and $\lambda_{\mathbf{z}_{t-1}}^{\kappa}$ is defined as $\frac{2}{1+\kappa|\mathbf{z}_{t-1}|^2}$.

Inspired by recent directional diffusion models [58, 59], we define the geodesic direction between the center of each cluster and the north pole $\mathbf{o}$ as the target diffusion direction, while imposing constraints to regulate the forward diffusion processes. In particular, the angular similarity constraints can be described as follows:

$$\mathbf{a}_u = \text{sgn}(\log_{\mathbf{o}}^{\kappa}(\mathbf{e}_{\mu_u})) * \epsilon_{\mathcal{B}}, \quad \mathbf{a}_i = \text{sgn}(\log_{\mathbf{o}}^{\kappa}(\mathbf{e}_{\mu_i})) * \epsilon_{\mathcal{B}}, \tag{12}$$

where $\mathbf{a}_u$ and $\mathbf{a}_i$ represent the angle constrained noise, $\mu$ is the clustering center corresponding to each user $u$ and item $i$. By integrating both radial and angular constraints, the geometric diffusion process (*cf.* Eq. (11)) can be reformulated as follows:

$$q(\mathbf{z}_t^u|\mathbf{z}_{t-1}^u) = \sqrt{1-\beta_t}\mathbf{z}_{t-1}^u + \sqrt{\beta_t}\mathbf{a}_u + \delta\tanh(\sqrt{\kappa}\lambda_{\mathbf{z}_{t-1}^u}^{\kappa}/r)\mathbf{z}_{t-1}^u,$$
$$q(\mathbf{z}_t^i|\mathbf{z}_{t-1}^i) = \sqrt{1-\beta_t}\mathbf{z}_{t-1}^i + \sqrt{\beta_t}\mathbf{a}_i + \delta\tanh(\sqrt{\kappa}\lambda_{\mathbf{z}_{t-1}^i}^{\kappa}/r)\mathbf{z}_{t-1}^i. \tag{13}$$

**THEOREM 2.** *Let $\mathbf{z}_t$ denotes the user or item at the $t$-step in the forward diffusion process Eq. (13). As $t \to \infty$, the low-dimensional latent representation $\mathbf{z}_t$ satisfies:*

$$\lim_{t\to\infty} \mathbf{z}_t \sim \mathcal{N}_f(\delta\mathbf{z}_{t-1}, \mathbf{I}), \tag{14}$$

*where $\mathcal{N}_f$ is an approximate folded normal distribution. More detail and proof can be referred to in the Appendix A.4.*

*3.2.3 Reverse Process.* After getting noisy user embeddings $\mathbf{z}_T^u$ and noisy item embeddings $\mathbf{z}_T^i$ in the forward process, we follow the standard denoising process (*cf.* Eq. (2)) and train a denoising network to simulate the process of reverse diffusion.

$$p_\theta(\hat{\mathbf{z}}_{t-1}^u|\hat{\mathbf{z}}_t^u) = \mathcal{N}_{\mathcal{B}}(\hat{\mathbf{z}}_{t-1}^u; \mu_\theta(\hat{\mathbf{z}}_t^u, t), \Sigma_\theta(\hat{\mathbf{z}}_t^u, t)),$$
$$p_\psi(\hat{\mathbf{z}}_{t-1}^i|\hat{\mathbf{z}}_t^i) = \mathcal{N}_{\mathcal{B}}(\hat{\mathbf{z}}_{t-1}^i; \mu_\psi(\hat{\mathbf{z}}_t^i, t), \Sigma_\psi(\hat{\mathbf{z}}_t^i, t)), \tag{15}$$

where $\hat{\mathbf{z}}_t^u$ and $\hat{\mathbf{z}}_t^i$ are the denoised embeddings in the reverse step $t$, $\theta$ and $\psi$ are the learnable parameters of the user denoising module and the item denoising module correspondingly. These denoising modules are executed iteratively in the reverse process until the generation of final clean embeddings $\hat{\mathbf{z}}_0^u$ and $\hat{\mathbf{z}}_0^i$.

## 3.3 Optimization

*3.3.1 Hyperbolic Margin-based Ranking Loss.* The margin-based ranking loss has shown to be quite beneficial for hyperbolic recommender methods [41, 57, 62]. This loss aims to distinguish user-item pairs up to a specified margin into positive and negative samples, once the margin is satisfied the pairs are regarded as well separated. Specifically, for each user $u$ we sample a positive item $i$ and a negative item $j$, and the margin loss is described as:

$$\mathcal{L}_{\textbf{Rec}}(u, i, j) = max(\underbrace{\mathbf{s}(u, j)}_{push} - \underbrace{\mathbf{s}(u, i)}_{pull} + m, 0), \tag{16}$$

where the $\mathbf{s}(\cdot)$ denotes the Fermi-Dirac decoder (*cf.* Eq. (9)), $m$ is the margin between $(u, i)$ and $(u, j)$. As a result, positive items are pulled closer to user while negative items are pushed outside the margin.

*3.3.2 Reconstruction Loss.* To improve the embedding denoising process, it is crucial to minimize the variational lower bound of the predicted user and item embeddings. Based on the KL divergence

derived from the multivariate Gaussian distribution (*cf.* Eq. (8)), the reconstruction loss of denoising process is stated as follows:

$$\mathcal{L}_{\text{re}}(u, i) = \mathbb{E}_q \left[ -\log p_\theta(\hat{\mathbf{z}}_0^u) - \log p_\psi(\hat{\mathbf{z}}_0^i) \right], \tag{17}$$

where $\hat{\mathbf{z}}_0^u$ and $\hat{\mathbf{z}}_0^i$ are derived from the final step of Eq. (15).

To reduce computational complexity, we follow paper [66] by uniformly sampling t from {1, 2, ..., T} and simplify Eq. (17) into the following equation:

$$\mathcal{L}_{\text{re}}(u, i) = (\mathcal{L}_{\text{re}}^u + \mathcal{L}_{\text{re}}^i)/2, \quad \text{where} \tag{18}$$

$$\mathcal{L}_{\text{re}}^u = \mathbb{E}_{t \sim \mathcal{U}(1,\mathrm{T})} \mathbb{E}_q \left[ ||z_0^u - \hat{\mathbf{z}}_0^u||_2^2 \right], \\ \mathcal{L}_{\text{re}}^i = \mathbb{E}_{t \sim \mathcal{U}(1,\mathrm{T})} \mathbb{E}_q \left[ ||z_0^i - \hat{\mathbf{z}}_0^i||_2^2 \right]. \tag{19}$$

*3.3.3 Total Loss.* The total loss function of HDRM comprises two parts: a hyperbolic margin-based ranking loss for recommendation, and a reconstruction loss for the denoising process. In summary, the total loss function of HDRM is formulated as follows:

$$\mathcal{L}(u, i, j) = \alpha \cdot \mathcal{L}_{\textbf{Rec}}(u, i, j) + (1 - \alpha) \cdot \mathcal{L}_{\text{re}}(u, i), \tag{20}$$

where $\alpha$ is a balance factor to adjust the weight of these two losses.

To further refine HDRM, we introduce a reweighted loss aimed at improving data cleaning. Drawing inspiration from the previous works [49, 66], we dynamically assign lower weights to instances with lower positive scores:

$$\text{w}(u, i, j) = \text{sigmoid}(\mathbf{s}(u, i))^\beta, \tag{21}$$

$$\mathcal{L}_{\text{total}}(u, i, j) = \text{w}(u, i, j)\mathcal{L}(u, i, j), \tag{22}$$

where $\beta$ is the reweighted factor which regulates the range of weights, $\mathbf{s}(u, i)$ is obtained from Eq. (9). Consequently, we redefine the total loss function of HDRM as presented in Eq. (22).

## 3.4 Complexity Analysis

*3.4.1 Time Complexity.* The time complexity of our model is primarily composed of two phases: 1) Hyperbolic embedding and clustering; 2) Diffusion forward process.

- **Hyperbolic embedding and clustering**: We encode each user and item into hyperbolic space using hyperbolic GCN. This process results in $n * d$-dimensional vectors, where n is the total number of users and items. The time complexity of this step is $O(nd) * 1(t)$, where $1(t)$ represents the time cost of passing through the neural network. The clustering process has an approximate time complexity of $O(cnd)$, where $c$ denotes the number of cluster categories.
- **Diffusion forward process**: For the forward process of diffusion, a single noise addition step suffices. This step has a time complexity of $O(nd)$. The training of denoising networks incurs a complexity of $O(nd) * 1(t)$.

In summary, the overall time complexity for each epoch is $O(1(t) * 2nd) + O((c + 1)nd)$.

*3.4.2 Space Complexity.* In HDRM, we encode users and items in hyperbolic space, representing each as an $n * d$-dimensional vectors. This encoding scheme results in a diffusion scale of $O(hnd)$, where $h$ denotes the total number of user-item interactions.

# 4 EXPERIMENTS

In this section, we conduct a series of experiments to validate HDRM and answer the following key research questions:

- **RQ1**: How does HDRM perform compared to baseline models on real-world datasets?
- **RQ2**: How does each proposed module contribute to the performance?
- **RQ3**: How does HDRM perform in mitigating the effects of noisy data?
- **RQ4**: How do hyper-parameters influence the performance of HDRM?

## 4.1 Experimental Settings

*4.1.1 Datasets and Evaluation Metrics.* We evaluate HDRM on three real-world datasets: Amazon-Book[1], Yelp2020[2], and ML-1M[3]. The detailed statistical information is presented in the Table 1. Across all datasets, interactions rating below 4 classify as false-positive engagements. We follow the data partition rubrics in recent collaborative filtering methods [35] [12] and split into three parts (training sets, validation sets, and test sets) with a ratio 7:1:2. Our evaluation of top-K recommendation efficiency involves the full-ranking protocol, incorporating two popular metrics Recall@K (R@K) and NDCG@K (N@K) for which we use K values of 10 and 20.

**Table 1: Statistics of three datasets under two different settings, where "C" and "N" represent clean training and natural noise training, respectively. "Int." denotes interactions.**

| Dataset | #User | #Item (C) | #Int. (C) | #Item (N) | #Int. (N) |
|---|---|---|---|---|---|
| Amazon-Book | 108,822 | 94,949 | 3,146,256 | 178,181 | 3,145,223 |
| Yelp2020 | 54,574 | 34,395 | 1,402,736 | 77,405 | 1,471,675 |
| ML-1M | 5,949 | 2,810 | 571,531 | 3,494 | 618,297 |

*4.1.2 Baselines and Hyper-parameter Settings.* The effectiveness of our method is assessed through comparison with the following baselines: classic collaborative filtering methods include BPRMF [35] and LightGCN [12]. Autoencoder-based recommender methods are represented by CDAE [54] and Multi-DAE [24]. Diffusion-based recommender methods include CODIGEM [47], DiffRec [23], and DDRM [66]. Finally, hyperbolic recommender methods encompass HyperML [46], HGCF [41], and HICF [57]. It is worth noting that the complete form of our adopted DDRM is LightGCN+DDRM. Further details on these models can be found in Appendix B.1.1. More details about our HDRM's hyper-parameter settings can be found in Appendix B.1.2.

## 4.2 Overall Performance Comparison (RQ1)

Table 2 reports the comprehensive performance of all the compared baselines across three datasets. Based on the results, the main observations are as follow:

---

[1] https://jmcauley.ucsd.edu/data/amazon/
[2] https://www.yelp.com/dataset/
[3] https://grouplens.org/datasets/movielens/1m/

**Table 2: The overall performance evaluation results for the proposed method and compared baseline models on three experimented datasets, highlighting the best and second-best performances in bold and borderline, respectively. Numbers with an asterisk (\*) indicate statistically significant improvements over the best baseline (t-test with p-value <0.05).**

| Model | ML-1M | | | | Amazon-Book | | | | Yelp2020 | | | |
|---|---|---|---|---|---|---|---|---|---|---|---|---|
| | R@10 | N@10 | R@20 | N@20 | R@10 | N@10 | R@20 | N@20 | R@10 | N@10 | R@20 | N@20 |
| BPRMF (UAI2009) | 0.0876 | 0.0749 | 0.1503 | 0.0966 | 0.0437 | 0.0264 | 0.0689 | 0.0339 | 0.0341 | 0.0210 | 0.0560 | 0.0276 |
| LightGCN (SIGIR2020) | 0.0987 | 0.0833 | 0.1707 | 0.1083 | 0.0534 | 0.0325 | 0.0822 | 0.0411 | 0.0540 | 0.0325 | 0.0904 | 0.0436 |
| CDAE (WSDM2016) | 0.0991 | 0.0829 | 0.1705 | 0.1078 | 0.0538 | 0.0361 | 0.0737 | 0.0422 | 0.0444 | 0.0280 | 0.0703 | 0.0360 |
| MultiDAE (WWW2018) | 0.0995 | 0.0803 | 0.1753 | 0.1067 | 0.0571 | 0.0357 | 0.0855 | 0.0422 | 0.0522 | 0.0316 | 0.0864 | 0.0419 |
| HyperML (WSDM2020) | 0.0997 | 0.0832 | 0.1752 | 0.1042 | 0.0567 | 0.0362 | 0.0846 | 0.0432 | 0.0539 | 0.0311 | 0.0911 | 0.0409 |
| HGCF(WWW2021) | 0.1009 | 0.0865 | 0.1771 | 0.1126 | 0.0633 | 0.0392 | 0.0931 | 0.0481 | 0.0560 | 0.0329 | 0.0931 | 0.0447 |
| HICF (KDD2022) | 0.9970 | 0.0848 | 0.1754 | 0.1010 | 0.0652 | 0.0426 | 0.0984 | 0.0514 | 0.0590 | 0.0366 | 0.0968 | 0.0488 |
| CODIGEM (KSEM2022) | 0.0972 | 0.0837 | 0.1699 | 0.1087 | 0.0300 | 0.0192 | 0.0478 | 0.0245 | 0.0470 | 0.0292 | 0.0775 | 0.0385 |
| DiffRec (SIGIR2023) | 0.1023 | 0.0876 | 0.1778 | 0.1136 | 0.0695 | 0.0451 | 0.1010 | 0.0547 | 0.0581 | 0.0363 | 0.0960 | 0.0478 |
| DDRM (SIGIR2024) | 0.1017 | 0.0874 | 0.1760 | 0.1132 | 0.0685 | 0.0432 | 0.0994 | 0.0521 | 0.0556 | 0.0343 | 0.0943 | 0.0438 |
| HDRM | **0.1089\*** | **0.0943\*** | **0.1859\*** | **0.1201\*** | **0.0732\*** | **0.0517\*** | **0.1065\*** | **0.0591\*** | **0.0630\*** | **0.0395\*** | **0.1034\*** | **0.0496\*** |
| Improv. | 6.5% | 7.6% | 4.6% | 5.7% | 5.3% | 14.6% | 5.4% | 8.0% | 6.8% | 7.9% | 6.8% | 1.6% |

**Table 3: Performance of different design variations on the three datasets. The bolded numbers denote the most significant change in performance.**

| Model | ML-1M | | | | Amazon-Book | | | | Yelp2020 | | | |
|---|---|---|---|---|---|---|---|---|---|---|---|---|
| | R@10 | N@10 | R@20 | N@20 | R@10 | N@10 | R@20 | N@20 | R@10 | N@10 | R@20 | N@20 |
| HDRM | 0.1089 | 0.0943 | 0.1859 | 0.1201 | 0.0732 | 0.0517 | 0.1065 | 0.0591 | 0.0630 | 0.0395 | 0.1034 | 0.0496 |
| HDRM w/o $\mathcal{H}_\kappa^n$ | 0.1061 | 0.0919 | 0.1798 | 0.1153 | 0.0703 | 0.0473 | 0.1033 | 0.0566 | 0.0593 | 0.0383 | 0.0983 | 0.0476 |
| HDRM w/o Geo | 0.1059 | 0.0907 | 0.1788 | 0.1141 | **0.0694** | **0.0442** | **0.1003** | **0.0534** | **0.0568** | **0.0359** | **0.0978** | **0.0453** |
| HDRM w/o Diff | **0.1041** | **0.0883** | **0.1775** | **0.1139** | 0.0695 | 0.0457 | 0.1012 | 0.0547 | 0.0589 | 0.0382 | 0.0976 | 0.0463 |

- Our proposed HDRM demonstrates consistent performance improvements across all metrics on three datasets compared to state-of-the-art baselines. This superior performance is primarily attributed to three key factors: 1) HDRM excels in capturing the complex relationships in user-item interactions compared to Euclidean-based approaches. This capability allows for a more nuanced understanding of the underlying recommendation dynamics. 2) By employing neural networks to incrementally learn each denoising transition step from $t$ to $t$-1, HDRM effectively models complex distributions. This approach significantly enhances the model's capacity to capture intricate patterns in the data. 3) Through learning the data distribution, HDRM exhibits superior capabilities in addressing data sparsity issues. This enables the model to infer latent associations from limited data.

- Diffusion-based approaches, such as DDRM and DiffRec, generally outperform traditional methods like BPRMF and LightGCN. This superior performance can be attributed to the alignment between their generative frameworks and the processes underlying user-item interactions. Among the generative methods, DiffRec demonstrates particularly impressive results, leveraging variational inference and KL divergence to achieve more robust generative modeling. In contrast, CODIGEM underperforms compared to LightGCN and other generative methods, primarily due to its reliance on only the first autoencoder for inference.

- Diffusion-based recommendation models do not universally outperform hyperbolic-based models. For instance, on the Yelp2020 dataset, HICF demonstrates superior performance compared to DiffRec. While diffusion-based models exhibit enhanced robustness and noise-handling capabilities, hyperbolic spaces are inherently well-suited for representing data with hierarchical structures and power-law distributions—characteristics that closely align with user-item interaction graphs in numerous recommender systems. Notably, models that integrate hyperbolic geometry with diffusion techniques have exhibited superior performance across three datasets by leveraging the strengths of both approaches.

### 4.3 Ablation Study (RQ2)

To validate the effectiveness of our proposed method, we conducted ablation studies by removing three key components from HDRM: the hyperbolic encoder (HDRM w/o $\mathcal{H}_\kappa^n$), geometric constraints (HDRM w/o Geo) and diffusion model (HDRM w/o Diff). Table 3 presents the results of our experiments on three datasets, from which we draw the following significant conclusions:

- The model's performance significantly decreases when the diffusion model, geometric constraints, and hyperbolic encoder are removed individually. This demonstrates the crucial role these modules play in the model's effectiveness. Furthermore, the table

**Table 4: Comparative analysis of best diffusion methods (DiffRec) and hyperbolic approaches (HICF) in noisy datasets, focusing on their performance amid random clicks and other data imperfections, highlighting the best and second-best performances in bold and borderline, respectively. Numbers with an asterisk (*) indicate statistically significant improvements over the best baseline (t-test with p-value <0.05).**

| Model | ML-1M | | | | Amazon-Book | | | | Yelp2020 | | | |
|---|---|---|---|---|---|---|---|---|---|---|---|---|
| | R@10 | N@10 | R@20 | N@20 | R@10 | N@10 | R@20 | N@20 | R@10 | N@10 | R@20 | N@20 |
| HICF (KDD2022) | 0.0635 | 0.0437 | 0.1211 | 0.0643 | 0.0512 | 0.0298 | 0.0763 | 0.0374 | 0.4770 | 0.0286 | 0.815 | 0.0387 |
| DiffRec (SIGIR2023) | 0.0658 | 0.0488 | 0.1236 | 0.0703 | 0.0537 | 0.0329 | 0.0806 | 0.0411 | 0.0501 | 0.0307 | 0.0847 | 0.0412 |
| DDRM (SIGIR2024) | 0.0667 | 0.0508 | 0.1221 | 0.0710 | 0.0468 | 0.0273 | 0.0742 | 0.0355 | 0.0516 | 0.0305 | 0.0870 | 0.0412 |
| HDRM | 0.0679* | 0.0522* | 0.1254* | 0.0714* | 0.0554* | 0.0336* | 0.0819* | 0.0427* | 0.0523* | 0.0325* | 0.0883* | 0.0432* |

3 reveals that the absence of the diffusion model and geometric constraints has a more substantial impact on the model's performance compared to the hyperbolic encoder. This discrepancy may be attributed to the inherent hierarchical structure and information-rich properties of hyperbolic space. However, without geometric constraints, the learned embeddings might become overly dispersed or concentrated within the space, failing to fully leverage the advantages of hyperbolic geometry. In contrast to the diffusion component, real-world recommendation models may rely more heavily on capturing the propagation and evolution of preferences rather than strictly adhering to hierarchical structures.

- The removal of the diffusion model results in the most significant performance decline on the ML-1M dataset, while the elimination of geometric constraints leads to the most substantial performance drop on the Amazon-Book and Yelp2020 datasets. This discrepancy may be attributed to the higher density of the ML-1M dataset compared to Amazon-Book and Yelp2020. The marked performance degradation observed when removing the diffusion model from the relatively dense ML-1M dataset underscores the critical role of the diffusion process in modeling complex and dynamic user behaviors. The higher density of ML-1M implies more frequent user-item interactions and intricate information flow compared to other datasets. In such an environment, diffusion models may more effectively capture rapidly evolving user preferences, social influences, and non-linear relationships.

In conclusion, our ablation studies highlight the significant contributions of each module in HDRM to the overall model performance. These findings not only validate our design choices but also provide insights into the relative importance of different components in hyperbolic recommender models.

## 4.4 Robustness Analysis (RQ3)

In real-world recommender systems, user behavior data often contains noise, such as random clicks or unintentional interactions. To evaluate HDRM's effectiveness in handling noisy data, we conducted a comparative analysis with DiffRec and DDRM, the leading diffusion methods, and HICF, the leading hyperbolic approach. Our noise comprises natural noise (*cf.* Table 1) and randomly sampled interactions, maintaining an equal scale for both components.

Table 4 presents the performance metrics of these models in the presence of noise. The results demonstrate that HDRM consistently outperforms both HICF, DDRM and DiffRec, validating

its robustness against noisy data. Notably, diffusion-based models exhibit superior performance in noisy environments, which aligns with theoretical expectations. This can be attributed to the inherent denoising process that underpins diffusion models, making them particularly well-suited for mitigating the impact of erroneous user interactions. In contrast, HICF's performance degraded significantly in the presence of noise, suggesting that the hyperbolic space does not offer a substantial advantage over Euclidean space in terms of reducing the influence of noisy interactions. This finding challenges the presumed benefits of hyperbolic embeddings in this context and highlights the need for further investigation into their limitations in noisy recommendation scenarios.

## 4.5 In-depth Analysis (RQ4)

*4.5.1 Diffusion Step Analysis.* We investigate the impact of varying diffusion and inference steps on HDRM's performance. Figure 3 illustrates our experimental results across three datasets, HDRM's performance initially improves as diffusion and inference steps increase. However, it subsequently declines with further increases in these steps. This phenomenon can be attributed to several factors. When the number of diffusion steps is insufficient, the model lacks adequate iterations to progressively refine recommendation results, leading to suboptimal capture of user preferences. Conversely, an excessive number of diffusion steps may cause the model to overfit the noise distribution, potentially discarding valuable information from the original data. Similarly, an insufficient number of inference steps prevents the model from fully recovering the original data distribution from a pure noise state. However, an excessive number of inference steps can result in over-optimization, potentially causing the model to deviate from the target distribution. More diffusion step results can be found in Appendix B.2.1.

*4.5.2 Margin Analysis.* We investigate the impact of varying margin values on HDRM's performance. Figure 4 presents the experimental results, revealing a non-monotonic relationship between margin size and HDRM's performance. As the margin increases, HDRM's performance initially improves before subsequently declining, indicating the existence of an optimal margin value for maximizing model effectiveness. On the ML-1M dataset, the model achieves peak performance at a margin of 0.1. In contrast, for the Amazon-Book and Yelp2020 datasets, optimal model performance is attained at a margin of 0.2. This discrepancy is notable across different datasets. The Amazon-Book and Yelp2020 datasets show

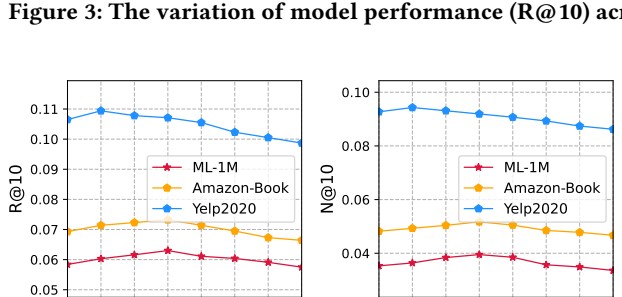

(a) ML-1M  (b) Amazon-Book  (c) Yelp2020

**Figure 3: The variation of model performance (R@10) across three datasets as diffusion steps and inference steps change.**

**Figure 4: The variation of model performance across three datasets as the margin changes.**

greater distinction between positive and negative samples than ML-1M. Considering the hyperbolic margin loss function, the margin represents the expected difference in scores between positive and negative samples. When dealing with datasets characterized by substantial disparities between positive and negative samples, a larger margin is advisable.

## 5 RELATED WORK

In this section, we review two relevant prior works: hyperbolic representation learning and generative recommendation.

### 5.1 Hyperbolic Representation Learning

Currently, Non-Euclidean representation learning, particularly hyperbolic representation learning, plays a crucial role in recommender systems (RSs)[41, 46, 65]. HyperML[46] investigates metric learning in hyperbolic space and its connection to collaborative filtering. Similarly, HGCF [41] proposes a hyperbolic GCN model for CF. In order to address the power-law distribution in recommender systems, HICF [57] focuses on enhancing the attention towards tail items in hyperbolic spaces, incorporating geometric awareness into the pull and push process. Interestingly, GDCF [65] aims to capture intent factors across geometric spaces by learning geometric disentangled representations associated with user intentions and different geometries. On the other hand, the paper [56] highlights that the naive inner product used in the factorization machine model [34]

may not adequately capture spurious or implicit feature interactions. Collaborative metric learning [16] proposes that learning the distance instead of relying on the inner product provides benefits in capturing detailed embedding spaces that encompass item-user interactions, item-item relationships, and user-user distances simultaneously. Consequently, the triangle inequality emerges as a more favorable alternative to the inner product.

### 5.2 Generative Recommendation

Generative models, such as Generative Adversarial Networks (GANs) [9, 18, 48] and Variational Autoencoders (VAEs) [24, 30, 63], play an important role in personalized recommendations but suffer from structural drawbacks [19, 38]. Recently, diffusion models have emerged as an alternative, offering better stability and representation capabilities, especially in recommendation systems [4, 17, 29, 52, 53]. Models like CODIGEM [47] and DiffRec [50] use diffusion models to predict user preferences by simulating interaction probabilities. Meanwhile, other approaches [7, 23, 25, 51] focus on content generation at the embedding level, similar to our DDRM framework. For instance, DiffRec [23] and CDDRec [51] add noise to target items in the forward process, later reconstructing them based on users' past interactions. DiffuASR [25] applies diffusion models to generate item sequences, addressing data sparsity challenges. Furthermore, DDRM [66] leverages diffusion models to denoise implicit feedback, leading to more robust representations in learning tasks.

## 6 CONCLUSION

Motivated by the promising results obtained from recent diffusion-based recommender models [21, 50, 66], we have decided to explore a more complex architecture. Building on the success of hyperbolic recommender methods [41, 46, 57, 61], we investigate that they hold great potential in addressing the non-Euclidean structural anisotropy of the underlying diffusion process in user-item interaction graphs. To this end, we propose HDRM model architecture, further mathematical proofs and experiments demonstrate the superiority of this method. We believe that this paper represents a milestone in hyperbolic diffusion models and offers a valuable baseline for future research in this field.

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

# A METHODS

## A.1 Hyperbolic Spaces

Here, we provide a comparison of geometric operations between the Poincaré ball manifold and the Lorentz manifold (hyperboloid manifold) as summarized in Table 5. It outlines the notation, geodesic distance, logarithmic map, exponential map, parallel transport, and the origin point for both manifolds, along with their respective mathematical formulations. This table serves to summarize the computational methods for these operations across the two different manifolds, highlighting their similarities and differences.

## A.2 Proof of Theorem 1

In hyperbolic geometry, four commonly used equivalent models are the Klein model, the Poincare disk model, the Lorentz model, and the Poincare half-plane model. For our analysis in this study, we utilize the Lorentz model.

Lorentz model $\mathbb{H}^n$ can be denoted by a set of points $z(z \in \mathbb{R}^{n+1})$ through Lorentzian product:

$$\langle z, z' \rangle_{\mathcal{L}} = -z_0 z_0' + \sum_{i=1}^{n} z_i z_i', \tag{23}$$

$$\mathbb{H}^n = \left\{ z \in \mathbb{R}^{n+1} : \langle z, z \rangle_{\mathcal{L}} = \frac{1}{\kappa}, z_0 > 0 \right\}. \tag{24}$$

Tangent space $T_{\boldsymbol{\mu}}\mathbb{H}^n$ is the tangent space of $\mathbb{H}^n$ at $\mu$. $T_{\boldsymbol{\mu}}\mathbb{H}^n$ can be represented as the set of points that satisfy the orthogonality relation with respect to the Lorentzian product:

$$T_{\boldsymbol{\mu}}\mathbb{H}^n := \{ \mathbf{u} : \langle \mathbf{u}, \boldsymbol{\mu} \rangle_{\mathcal{L}} = 0 \}. \tag{25}$$

*A.2.1 Parallel transport and inverse parallel transport.* For an arbitrary pair of point $\mu, \nu \in \mathbb{H}^n$, the parallel transport from $\nu$ to $\mu$ is defined as a map $\text{PT}_{\nu \to \mu}$ from $T_{\boldsymbol{\nu}}\mathbb{H}^n$ to $T_{\boldsymbol{\mu}}\mathbb{H}^n$ that carries a vector in $T_{\boldsymbol{\nu}}\mathbb{H}^n$ along the geodesic from $\nu$ to $\mu$ without changing its metric tensor.

The explicit formula for the parallel transport on the Lorentz model is given by:

$$\text{PT}_{\nu \to \mu}^{\kappa}(v) = v + \frac{\langle \mu - \alpha \nu, v \rangle_{\mathcal{L}}}{\alpha + 1}(v + \mu), \tag{26}$$

where $\alpha = -\langle v, \mu \rangle_{\mathcal{L}}$. The inverse parallel transport is given by:

$$v = \text{PT}_{\mu \to \nu}^{\kappa}(u). \tag{27}$$

*A.2.2 Exponential map and Logarithmic map.* Exponential map $\exp_m u : T_{\nu}\mathbb{H}^n \to \mathbb{H}^n$ is a map that we can use to project a vector $v \in T_{\boldsymbol{\mu}}\mathbb{H}^n$ to $\mathbb{H}^n$. The explicit formula for the exponential map on the Lorentz model is given by:

$$z = exp_{\boldsymbol{\mu}}^{\kappa}(\mathbf{u}) = \cosh(\|\mathbf{u}\|_{\mathcal{L}})\boldsymbol{\mu} + \sinh(\|\mathbf{u}\|_{\mathcal{L}})\frac{\mathbf{u}}{\|\mathbf{u}\|_{\mathcal{L}}}. \tag{28}$$

The logarithmic map is defined to compute the inverse of the exponential map, mapping the point back to the tangent space. It is given by:

$$u = \log_{\boldsymbol{\mu}}^{\kappa}(z) = \frac{\text{arccosh}(\alpha)}{\sqrt{\alpha^2 - 1}}(z - \alpha\boldsymbol{\mu}), \tag{29}$$

where $\alpha = -\langle \mu, z \rangle_{\mathcal{L}}$.

*A.2.3 Klein model.* This model of hyperbolic space is a subset of $R^n$ given by $K^n$, and a point in the Klein model can be obtained from the corresponding point in the hyperbolic model by projection:

$$\pi_{\mathbb{H} \to \mathbb{K}}(\mathbf{x})_i = \frac{x_i}{x_0}. \tag{30}$$

With its inverse given by:

$$\pi_{\mathbb{K} \to \mathbb{H}}^{-1}(\mathbf{x}) = \frac{1}{\sqrt{1 - \|\mathbf{x}\|^2}}(1, \mathbf{x}). \tag{31}$$

An interesting approach is that we can use the angle-preserving nature of the Klein model to construct a mapping from the Lorentz tangent plane to the Klein model via the spherical pole mapping $\text{P}^{\kappa}$:

$$k_j = \text{P}^{\kappa}(x_j) = \frac{1}{-\kappa + \sum_{i=1}^{n} x_i^2}(x_j^2). \tag{32}$$

*A.2.4 Proof.* In our model, $m$ represents the number of clusters and $\mathbf{e}_i (\mathbf{e}_i \in \mathbb{H}^n)$ represents the points in class $i (i \in 1, 2, 3, ..., m)$. $\mu_i$ denotes the hyperbolic center of the cluster $i$. Each point $\mathbf{e}_i$ will be mapped to the tangent plane $T_{\boldsymbol{\mu}_i}\mathbb{H}^n$ of its center $\mu_i$. Let $x_i$ denote the point after mapping to the tangent plane, it can be calculated by:

$$x_i = \log_{\boldsymbol{\mu}_i}^{\kappa}(\mathbf{e}_i). \tag{33}$$

If we consider all the points as being in the tangent plane to the North Pole $T_o\mathbb{H}^n$, then their corresponding coordinates are:

$$X = (x_1, x_2, ..., x_m). \tag{34}$$

For a curvature $\kappa_i$, if the following equation is satisfied:

$$\log_{\mathbf{o}}^{\kappa_i}(\mathbf{e}_i) = \text{PT}_{o \to \mu_i}^{\kappa_o}(\log_{\mathbf{o}}^{\kappa_o}(\mathbf{e}_i)), \tag{35}$$

then it is possible to transform the tangent planes from the various centers to the tangent plane at the North Pole and unify them into the Klein model. The mapping point $k$ is given by:

$$\begin{aligned} k_i &= \text{P}^{\kappa_o}(\log_{\mathbf{o}}^{\kappa_o}(\mathbf{e}_i)) \\ &= \text{P}^{\kappa_o}(\text{PT}_{\mu_i \to o}^{\kappa_o}(\log_{\boldsymbol{\mu}_i}(\mathbf{e}_i))) \\ &= \text{P}^{\kappa_o}(\text{PT}_{\mu_i \to o}^{\kappa_o}(x_i)) \\ &= \pi_{\mathbb{H} \to \mathbb{K}}(\mathbf{e}_i) \\ &= \text{P}^{\kappa_i}(\log_{o}^{\kappa_i}(\mathbf{e}_i)). \end{aligned} \tag{36}$$

This implies that our approach essentially involves mathematically projecting points to approximate a Klein model comprising multiple curvatures. The process represents a topological reconstruction of the geometric space derived from the original graph structure, thereby enhancing our ability to capture the geometric properties inherent in the original user-item interaction graph.

## A.3 Proof of the lack of additivity in the package normal distribution

*A.3.1 Description of symbols.* $\mathcal{B}_{\kappa}^d$ is a d-dimensional Poincaré Ball space with curvature $\kappa$. $\mathbf{R}^d$ is a d-dimensional Euclidean space. $\mathbf{G}(\mathbf{z})$ is the metric tensor of the hyperbolic space. $\mathbf{d}_{\mathbf{p}}^{\kappa}$ is the hyperbolic distance. We first introduce the expression of metric tensor $\mathbf{G}(\mathbf{z})$ in hyperbolic space and the hyperbolic distance $\mathbf{d}_{\mathbf{p}}^{\kappa}$:

$$\mathbf{G}(\mathbf{z}) = \begin{pmatrix} 1 & 0 \\ 0 & \left(\frac{\sinh(\sqrt{\kappa}r)}{\sqrt{\kappa}r}\right)^2 I_{d-1} \end{pmatrix}, \tag{37}$$

**Table 5: Summary of operations in the Poincaré ball manifold and the Lorentz manifold ($\kappa < 0$)**

| | Poincaré Ball Manifold | Lorentz Manifold (Hyperboloid Manifold) |
|---|---|---|
| **Notation** | $\mathcal{B}_\kappa^n = \left\{ \mathbf{x} \in \mathbb{R}^n : \langle \mathbf{x}, \mathbf{x} \rangle_2 < -\frac{1}{\kappa} \right\}$ | $\mathcal{L}_\kappa^n = \left\{ \mathbf{x} \in \mathbb{R}^{n+1} : \langle \mathbf{x}, \mathbf{x} \rangle_{\mathcal{L}} = \frac{1}{\kappa} \right\}$ |
| **Geodesics distance** | $d_{\mathcal{B}}^\kappa(\mathbf{x}, \mathbf{y}) = \frac{1}{\sqrt{|\kappa|}} \cosh^{-1}\left( 1 - \frac{2\kappa \|\mathbf{x}-\mathbf{y}\|_2^2}{(1+\kappa\|\mathbf{x}\|_2^2)(1+\kappa\|\mathbf{y}\|_2^2)} \right)$ | $d_{\mathcal{L}}^\kappa(\mathbf{x}, \mathbf{y}) = \frac{1}{\sqrt{|\kappa|}} \cosh^{-1}(\kappa \langle \mathbf{x}, \mathbf{y} \rangle_{\mathcal{L}})$ |
| **Logarithmic map** | $\log_{\mathbf{x}}^\kappa(\mathbf{y}) = \frac{2}{\sqrt{|\kappa|}\lambda_{\mathbf{x}}^\kappa} \tanh^{-1}\left( \sqrt{|\kappa|} \| - \mathbf{x} \oplus_\kappa \mathbf{y} \|_2 \right) \frac{-\mathbf{x} \oplus_\kappa \mathbf{y}}{\|-\mathbf{x} \oplus_\kappa \mathbf{y}\|_2}$ | $\log_{\mathbf{x}}^\kappa(\mathbf{y}) = \frac{\cosh^{-1}(\langle \mathbf{x}, \mathbf{y} \rangle_{\mathcal{L}})}{\sinh\left( \cosh^{-1}(\langle \mathbf{x}, \mathbf{y} \rangle_{\mathcal{L}}) \right)} (\mathbf{y} - \kappa \langle \mathbf{x}, \mathbf{y} \rangle_{\mathcal{L}} \mathbf{x})$ |
| **Exponential map** | $\exp_{\mathbf{x}}^\kappa(\mathbf{v}) = \mathbf{x} \oplus_\kappa \left( \tanh\left( \sqrt{|\kappa|} \frac{\lambda_{\mathbf{x}}^\kappa \|\mathbf{v}\|_2}{2} \right) \frac{\mathbf{v}}{\sqrt{|\kappa|}\|\mathbf{v}\|_2} \right)$ | $\exp_{\mathbf{x}}^\kappa(\mathbf{v}) = \cosh\left( \sqrt{|\kappa|} \|\mathbf{v}\|_{\mathcal{L}} \right) \mathbf{x}$ |
| **Parallel transport** | $P\mathcal{T}_{\mathbf{x} \to \mathbf{y}}^\kappa(\mathbf{v}) = \frac{\lambda_{\mathbf{x}}^\kappa}{\lambda_{\mathbf{y}}^\kappa} \text{gyr}[\mathbf{y}, -\mathbf{x}]\mathbf{v}$ | $P\mathcal{T}_{\mathbf{x} \to \mathbf{y}}^\kappa(\mathbf{v}) = \mathbf{v} - \frac{\kappa \langle \mathbf{y}, \mathbf{v} \rangle_{\mathcal{L}}}{1+\kappa \langle \mathbf{x}, \mathbf{y} \rangle_{\mathcal{L}}} (\mathbf{x} + \mathbf{y})$ |
| **Origin point** | $\mathbf{0}_n$ | $\left[ \frac{1}{\sqrt{|\kappa|}}, \mathbf{0}_n \right]$ |

$$\mathbf{d}_{\mathbf{p}}^\kappa(z, y) = \frac{1}{\sqrt{\kappa}} \cosh^{-1}\left( 1 + 2\kappa \frac{\|z-y\|^2}{(1-\kappa\|z\|^2)(1-\kappa\|y\|^2)} \right). \quad (38)$$

Following the $\mathcal{P}$-VAE (Nickel & Kiela, 2017) we can subsequently derive the differential and integral operators. This derivation is accomplished by applying transformations using hyperbolic polar coordinates and within the framework of Euclidean space. The resulting operators will be expressed in terms of these transformed coordinate systems, providing a different perspective on the mathematical operations in the given context:

$$ds_{\mathcal{B}_\kappa^d}^2 = (\lambda_z^\kappa)^2(dz_1^2 + \cdots + dz_d^2) = \frac{4}{(1-\kappa\|x\|^2)^2} dz^2$$
$$= \frac{4}{(1-\kappa\rho^2)^2}(d\rho^2 + \rho^2 ds_{S^{d-1}}^2), \quad (39)$$

let $\mathbf{r} = \mathbf{d}_{\mathbf{p}}^\kappa$, we have

$$r = \int_0^\rho \lambda_t^\kappa dt = \int_0^\rho \frac{2}{1-\kappa t^2} dt = \int_0^{\sqrt{\kappa}\rho} \frac{2}{1-t^2} \frac{dt}{\sqrt{\kappa}}$$
$$= \frac{2}{\sqrt{\kappa}} \tanh^{-1}(\sqrt{\kappa}\rho). \quad (40)$$

Then, following from our previous derivation, we can now establish the subsequent mathematical relationship. This relationship is formally expressed by the equation presented below:

$$ds_{\mathcal{B}_\kappa}^2 = \frac{4}{(1-\kappa\rho^2)^2} \frac{1}{4}(1-\kappa\rho^2)^2 dr^2 + \left( 2\frac{\rho}{1-\kappa\rho^2} \right)^2 ds_{S^{d-1}}^2$$

$$= dr^2 + \left( 2\frac{\frac{1}{\sqrt{\kappa}}\tanh(\sqrt{\kappa}\frac{r}{2})}{1-\kappa\left( \frac{1}{\sqrt{\kappa}}\tanh(\sqrt{\kappa}\frac{r}{2}) \right)^2} \right)^2 ds_{S^{d-1}}^2 \quad (41)$$

$$= dr^2 + \left( \frac{1}{\sqrt{\kappa}} \sinh(\sqrt{\kappa}r) \right)^2 ds_{S^{d-1}}^2,$$

when $\kappa \to 0$, through this process, we are able to derive and reconstruct the standard Euclidean line element, which is a fundamental concept in geometry. This line element can be expressed mathematically as:

$$\mathbf{ds}_{\mathbf{R}^d}^2 = \mathbf{dr}^2 + \mathbf{r}^2 \mathbf{ds}_{S^{d-1}}^2. \quad (42)$$

Then, we proceed to the next step in our mathematical analysis, which involves performing the integral calculation. This crucial part of the process can be expressed as follows

$$\int_{\mathcal{B}_\kappa^d} f(z) d\mathcal{M}(z) = \int_{\mathcal{B}_\kappa^d} f(z)\sqrt{|G(z)|}dz$$
$$= \int_{\mathcal{T}_\mu \mathcal{B}_\kappa^d \cong \mathcal{R}^d} f(\mathbf{v})\sqrt{|G(\mathbf{v})|}d\mathbf{v}$$
$$= \int_{\mathbb{R}_+} \int_{\mathbb{S}^{d-1}} f(r)\sqrt{|G(r)|}dr r^{d-1} ds_{S^{d-1}}$$
$$= \int_{\mathbb{R}_+} \int_{\mathbb{S}^{d-1}} f(r) \left( \frac{\sinh(\sqrt{\kappa}r)}{\sqrt{\kappa}r} \right)^{d-1} dr r^{d-1} ds_{S^{d-1}}$$
$$= \int_{\mathbb{R}_+} \int_{\mathbb{S}^{d-1}} f(r) \left( \frac{\sinh(\sqrt{\kappa}r)}{\sqrt{\kappa}} \right)^{d-1} dr ds_{S^{d-1}}. \quad (43)$$

*A.3.2 Proof.* We introduce the concept of the probability density distribution for the wrapped normal distribution, a circular probability distribution that arises from wrapping a normal distribution around a circle. To adapt this distribution to a hyperbolic space, we employ a mapping technique. This mapping process begins by considering the normal distribution in the tangent plane $T_\mu \mathcal{B}_\kappa^d$, which is a flat space that touches the hyperbolic space at a single point.

We then use the exponential map, a mathematical function that projects points from this tangent plane onto the curved surface of the hyperbolic space. This projection effectively wraps the normal distribution onto the hyperbolic geometry. To generate samples from this mapped distribution, one can follow these steps:

$$\mathbf{z} = \exp_\mu^\kappa\left( G(\mu)^{-\frac{1}{2}} v \right) = \exp_\mu^\kappa\left( \frac{v}{\lambda_\mu^\kappa} \right), \text{with } v \sim \mathcal{N}(\cdot|\mathbf{0}, \Sigma). \quad (44)$$

*A.3.3 Anisotropic.* In anisotropic environments or settings, where properties vary depending on direction, the probability density of the phenomenon in question can be mathematically expressed using the following equation:

$$\mathcal{N}_{\mathcal{B}_\kappa^d}^{\text{W}}(z|\mu, \Sigma) = \mathcal{N}\left( G(\mu)^{1/2} \log_\mu(z) \Big| \mathbf{0}, \Sigma \right) \left( \frac{\sqrt{\kappa}d_p^\kappa(\mu, z)}{\sinh(\sqrt{\kappa}d_p^\kappa(\mu, z))} \right)^{d-1}$$

$$= \mathcal{N}\left( \lambda_\mu^\kappa \log_\mu(z) \Big| \mathbf{0}, \Sigma \right) \left( \frac{\sqrt{\kappa}d_p^\kappa(\mu, z)}{\sinh(\sqrt{\kappa}d_p^\kappa(\mu, z))} \right)^{d-1}. \quad (45)$$

We can plug its density with introducing the variable $v = r\alpha = \lambda_\mu^\kappa \log_\mu(z)$ and utilizing the metric tensor, and we have

$$
\int_{\mathcal{B}_\kappa^d} \mathcal{N}_{\mathcal{B}_\kappa^d}^{\mathbf{W}}(z|\mu,\Sigma) d\mathcal{M}(z)
$$

$$
= \int_{\mathcal{T}_\mu \mathcal{B}_\kappa^d \cong \mathbb{R}^d} \mathcal{N}(v|\mathbf{0},\Sigma) \left(\frac{\sqrt{\kappa}\|v\|_2}{\sinh(\sqrt{\kappa}\|v\|_2)}\right)^{d-1} \sqrt{|G(v)|}\, dv
$$

$$
= \int_{\mathbb{R}^d} \mathcal{N}(v|\mathbf{0},\Sigma) \left(\frac{\sqrt{\kappa}\|v\|_2}{\sinh(\sqrt{\kappa}\|v\|_2)}\right)^{d-1} \left(\frac{\sinh(\sqrt{\kappa}\|v\|_2)}{\sqrt{\kappa}\|v\|_2}\right)^{d-1}\, dv
$$

$$
= \int_{\mathbb{R}^d} \mathcal{N}(v|\mathbf{0},\Sigma)\, dv.
$$

Next, we derive whether the sum of two independent wrapped normally distributed variables still satisfies the wrapped normal distribution:

$$
\mathcal{N}_{\mathcal{B}_\kappa^d}^{\mathbf{W}}(z_1|\mu_1,\Sigma_1) * \mathcal{N}_{\mathcal{B}_\kappa^d}^{\mathbf{W}}(z_2|\mu_2,\Sigma_2)
$$

$$
= \int_{\mathcal{B}_\kappa^d} \mathcal{N}_{\mathcal{B}_\kappa^d}^{\mathbf{W}}(z - z_2|\mu_1,\Sigma_1) \mathcal{N}_{\mathcal{B}_\kappa^d}^{\mathbf{W}}(z_2|\mu_2,\Sigma_2) d\mathcal{M}(z_2)
$$

$$
= \int_{\mathbb{R}^d} \mathcal{N}(v - v_2|\mathbf{0},\Sigma_1) \mathcal{N}(v_2|\mathbf{0},\Sigma_2) \left(\frac{\sqrt{\kappa}\|v - v_2\|_2}{\sinh(\sqrt{\kappa}\|v - v_2\|_2)}\right)^{d-1} dv_2
$$

$$
\mathcal{N}_{\mathcal{B}_\kappa^d}^{\mathbf{W}}(z_1|\mu_1,\Sigma_1) * \mathcal{N}_{\mathcal{B}_\kappa^d}^{\mathbf{W}}(z_2|\mu_2,\Sigma_2) \nrightarrow \mathcal{N}_{\mathcal{B}_\kappa^d}^{\mathbf{W}}(z|\mu,\Sigma). \tag{46}
$$

*A.3.4 Isotropic.* In the isotropic setting, the density of the wrapped normal is given by:

$$
\mathcal{N}_{\mathcal{B}_\kappa}^{\mathbf{W}}(z|\boldsymbol{\mu},\sigma^2) = \frac{dv^{\mathbf{W}}(z|\boldsymbol{\mu},\sigma^2)}{d\mathcal{M}(z)}
$$

$$
= (2\pi\sigma^2)^{-d/2} \exp\left(-\frac{d_p^\kappa(\boldsymbol{\mu},z)^2}{2\sigma^2}\right) \left(\frac{\sqrt{\kappa}d_p^\kappa(\boldsymbol{\mu},z)}{\sinh(\sqrt{\kappa}d_p^\kappa(\boldsymbol{\mu},z))}\right)^{d-1}. \tag{47}
$$

Its integral form can be given by:

$$
\int_{\mathcal{B}_\kappa^d} \mathcal{N}_{E_\kappa^d}^{\mathbf{W}}(z|\boldsymbol{\mu},\sigma^2) d\mathcal{M}(z)
$$

$$
= \int_{R_+} \int_{S^{d-1}} \frac{1}{Z^R} e^{-\frac{r^2}{2\sigma^2}} r^{d-1} dr ds_{S^{d-1}}, \tag{48}
$$

where $Z^R$ is the constant, and it is defined as

$$
Z^R = \zeta \binom{d-1}{k} e^{\frac{(d-1-2k)^2}{2}c\sigma^2} \left[1 + \mathrm{erf}\left(\frac{(d-1-2k)\sqrt{c}\sigma}{\sqrt{2}}\right)\right], \tag{49}
$$

$$
\zeta = \frac{2\pi^{d/2}}{\Gamma(d/2)} \sqrt{\frac{\pi}{2}} \sigma \frac{1}{(2\sqrt{c})^{d-1}} \sum_{k=0}^{d-1} (-1)^k.
$$

Thus, we can derive its additivity:

$$
\mathcal{N}_{\mathcal{B}_\kappa^d}^{\mathbf{W}}(z_1|\mu_1,\Sigma_1) * \mathcal{N}_{\mathcal{B}_\kappa^d}^{\mathbf{W}}(z_2|\mu_2,\Sigma_2)
$$

$$
= \int_{\mathcal{B}_\kappa^d} \mathcal{N}_{\mathcal{B}_\kappa^d}^{\mathbf{W}}(z - z_2|\mu_1,\Sigma_1) \mathcal{N}_{\mathcal{B}_\kappa^d}^{\mathbf{W}}(z_2|\mu_2,\Sigma_2) d\mathcal{M}(z_2)
$$

$$
= \int_{R_+} \int_{S^{d-1}} \frac{1}{Z^{R^2}} e^{-\frac{(r-r_2)^2}{2\sigma^2}} (r - r_2)^{d-1} \gamma_p^\kappa e^{-\frac{(r_2)^2}{2\sigma^2}} (r_2)^{d-1} dr ds_{S^{d-1}}
$$

$$
= \int_{R_+} \int_{S^{d-1}} \frac{1}{Z^{R^2}} e^{-\frac{(r^2-2rr_2+2r_2^2)}{2\sigma^2}} (r_2(r - r_2))^{d-1} \gamma_p^\kappa dr ds_{S^{d-1}}
$$

$$
\gamma_p^\kappa = \left(\frac{\sqrt{\kappa}d_p^\kappa(\boldsymbol{\mu}_1,z_1)}{\sinh(\sqrt{\kappa}d_p^\kappa(\boldsymbol{\mu}_1,z_1))}\right)^{d-1}
$$

$$
\mathcal{N}_{\mathcal{B}_\kappa^d}^{\mathbf{W}}(z_1|\mu_1,\Sigma_1) * \mathcal{N}_{\mathcal{B}_\kappa^d}^{\mathbf{W}}(z_2|\mu_2,\Sigma_2) \nrightarrow \mathcal{N}_{\mathcal{B}_\kappa^d}^{\mathbf{W}}(z|\mu,\Sigma). \tag{50}
$$

Thus, we have demonstrated the lack of additivity in the wrapped normal distribution.

## A.4 Proof of Theorem 2

First, we derive the form of the probability distribution of the forward diffusion process.

*A.4.1 Definition: The folded normal distribution.* If the probability distribution of $Y$ follows the Gaussian distribution, with $Y \sim N(\mu,\sigma^2)$, thus, $Z = |Y|$, satisfies $Z \sim \mathcal{N}_F(\mu,\sigma^2)$, where $\mathcal{N}_F(\mu,\sigma^2)$ denotes the folded normal distribution with mean $\mu$ and variance $\sigma$. The density of $Z$ is given by

$$
f(\mathbf{z}) = \frac{1}{\sqrt{2\pi\sigma^2}} \left[ e^{-\frac{1}{2\sigma^2}(\mathbf{z}-\mu)^2} + e^{-\frac{1}{2\sigma^2}(\mathbf{z}+\mu)^2} \right]. \tag{51}
$$

The density can be written in a more attractive form

$$
f(\mathbf{z}) = \sqrt{\frac{2}{\pi\sigma^2}} e^{-\frac{(\mathbf{z}^2+\mu^2)^2}{2\sigma^2}} \cosh\left(\frac{\mu\mathbf{z}}{\sigma^2}\right). \tag{52}
$$

Specifically, when $\mu = 0$, the density can be represented by

$$
f(\mathbf{z}) = \sqrt{\frac{2}{\pi\sigma^2}} e^{-\frac{\mathbf{z}^2}{2\sigma^2}}, \tag{53}
$$

which is also named the half-normal distribution.

*A.4.2 Definition.* The random variable $Z$ obeys the probability distribution $N_f(\mu,\sigma)$ if and only if the density of $Z$ can be given by

$$
f(\mathbf{z}) = \begin{cases} \sqrt{\frac{2}{\pi\sigma^2}} e^{-\frac{(\mathbf{z}-\mu)^2}{2\sigma^2}}, & \mathbf{z} \geq \mu \\ 0, & \mathbf{z} < \mu \end{cases} \tag{54}
$$

*A.4.3 Proof.* Now, we can prove this Theorem, the angle-constrained noise $n$ in the forward diffusion process is given by Eq. (12). For the convenience of the later derivation, it can be assumed that $\mathrm{sgn}(\mathrm{logmap}_o^\kappa(\mathbf{e}_m)) = 1$ Thus, according to the definition of the folded normal distribution, it follows that:

$$
n \sim \mathcal{N}_F(0,I), \tag{55}
$$

Similarly, it can be easily obtained from Eq. (13) and Eq. (55) that the density of $\mathbf{z}_t$ in the diffusion process can be written by:

$$
f(\mathbf{z}_t) = \begin{cases} \sqrt{\frac{2}{\pi\sigma_t^2}} e^{-\frac{(\mathbf{z}-\mu_t\mathbf{z}_0)^2}{2\sigma_t^2}}, & \mathbf{z} \geq \mu_t\mathbf{z}_0 \\ 0, & \mathbf{z} < \mu_t\mathbf{z}_0 \end{cases} \tag{56}
$$

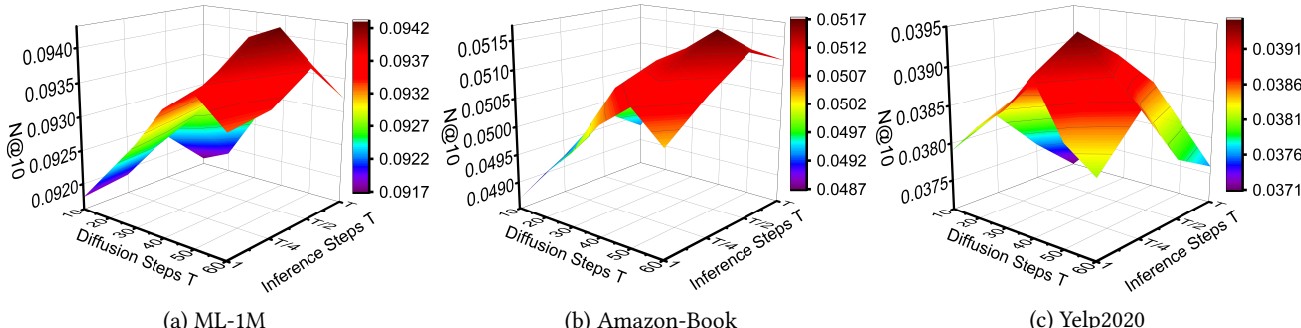

(a) ML-1M

(b) Amazon-Book

(c) Yelp2020

**Figure 5: The variation of model performance across three datasets as diffusion steps and inference steps change.**

where $\mu_t = \sqrt{\overline{\alpha}_t} + \delta \tanh[\sqrt{\kappa}\lambda_o^\kappa(t)/T_0]$, and $\sigma_t = (1 - \overline{\alpha}_t)I$.

Thus, the probability density distribution of $\mathbf{z}_t$ satisfies:

$$p(\mathbf{z}_t|\mathbf{z}_0) = N_f(\mu_t, \sigma_t). \tag{57}$$

and

$$\lim_{t \to \infty} \mathbf{z}_t \sim N_f(\delta\mathbf{z}_0, I). \tag{58}$$

# B EXPERIMENTS

## B.1 Experimental Settings

*B.1.1 Baselines.* The detailed information of the baselines is as follows:

**Classic Collaborative Filtering Methods:**

- **BPRMF** [35]: This is a typical collaborative filtering method that optimizes MF with a pairwise ranking loss.
- **LightGCN** [12]: This is an effective GCN-based collaborative filtering method, which improves performance by eliminating non-linear projection and activation.

**Auto-Encoders Recommender Methods:**

- **CDAE** [54]: This is a collaborative filtering method that applies denoising auto-encoders with user-specific latent factors to improve top-N recommendation performance.
- **MultiDAE** [24]: This is a variational autoencoder approach with partial regularization and multinomial likelihood for collaborative filtering on implicit feedback data

**Diffusion Recommender Methods:**

- **CODIGEM** [47]: This method employs a simple CL approach that avoids graph augmentations and introduces uniform noise into the embedding space to generate contrastive views.
- **DiffRec** [50]: This method uses LightGCN as the backbone and incorporates a series of structural augmentations to enhance representation learning.
- **DDRM** [66]: This is a plug-in denoising diffusion model that enhances robust representation learning for existing recommender systems by iteratively injecting and removing noise from user and item embeddings.

**Hyperbolic Recommender Methods:**

- **HyperML** [46]: This method is the first to propose using hyperbolic margin ranking loss for predicting user preferences toward items.

- **HGCF** [41]: This method is the first hyperbolic GCN model for collaborative filtering that can be effectively learned using a margin ranking loss.
- **HICF** [57]: This method adapts hyperbolic margin ranking learning by making the pull and push procedures geometric-aware, aiming to provide informative guidance for the learning of both head and tail items.

*B.1.2 Hyper-parameter Settings.* We determine the optimal hyper-parameters based on the Recall@20 metric evaluated on the validation set. For our Hyperbolic model, we tune these key parameters. The learning rate is varied among $\{1e^{-4}, 5e^{-4}, 1e^{-3}, 5e^{-3}\}$, while the curvature c is set to either -1 or 1. We explore GCN architectures with $\{2, 3, 4\}$ layers, and weight decay values of $\{0.001, 0.005, 0.01\}$. The margin is tested at $\{0.05, 0.1, 0.15, 0.2, 0.25, 0.3, 0.35, 0.4\}$. For the diffusion model, we investigate diffusion steps $T$ ranging from $\{10, 20, 30, 40, 50, 60\}$. The noise schedule is bounded between $1e^{-4}$ and $1e^{-2}$. We explore loss balance factors $\lambda$ from $\{0.1, 0.2, ..., 0.6\}$, and denoising weight factors $\gamma$ from $\{0, 0.05, 0.1, 0.2, ..., 0.9\}$. All experiments are conducted using PyTorch on a server equipped with 16 Intel Xeon CPUs @2.10GHz and an NVIDIA RTX 4090 GPU, ensuring efficient training and evaluation of our models across this extensive hyperparameter space.

## B.2 More Experimental Results

*B.2.1 Diffusion Step Analysis.* Here is further analysis of the diffusion steps. Figure 5 illustrates how the model performance metric N@10 changes across three datasets as the diffusion steps and inference steps vary.

*B.2.2 Embedding Visualization.* Figures 6, 7, and 8 present t-SNE visualizations of item embeddings learned by DDRM, HICF, and HDRM on the ML-1M, Amazon-Book, and Yelp2020 datasets, offering insights into our model's capability to address distribution shifts. We categorize items based on their popularity in the training set. For ML-1M and Yelp2020, the top 50% most popular items are designated as "popular", while the bottom 50% are labeled "unpopular". Due to its larger size, the Amazon-Book dataset uses a 20-80 split for popular and unpopular items, respectively.

The visualizations reveal that DDRM's learned embeddings for popular and unpopular items maintain a noticeable separation in

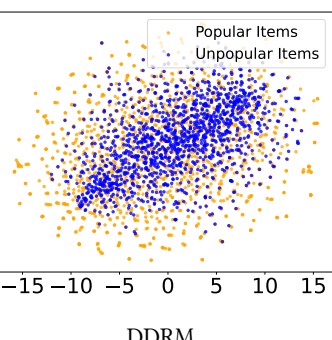 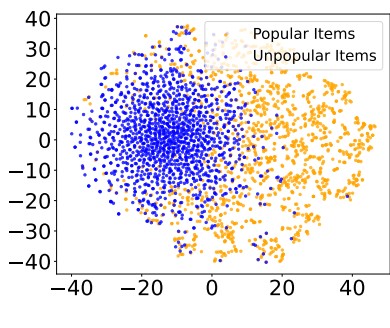 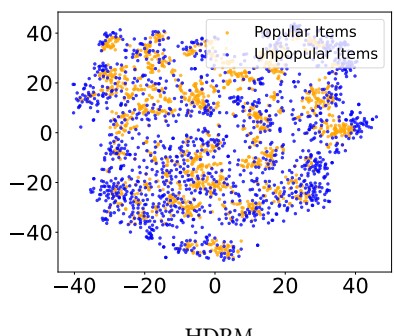

DDRM · HICF · HDRM

Figure 6: Visualize the distribution of item embeddings on the ML-1M dataset using DDRM, HICF, and HDRM. HDRM ensures that popular and unpopular items have representations with almost the same positions in the same space. .

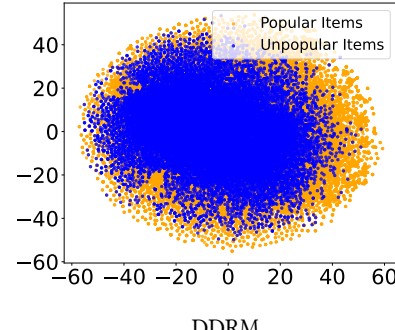 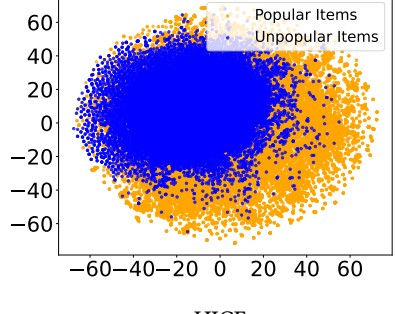 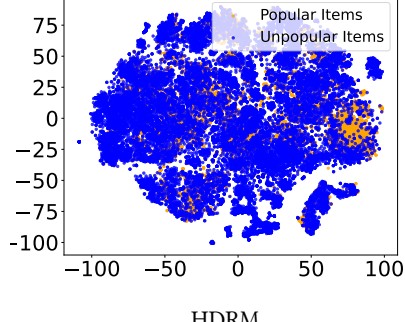

DDRM · HICF · HDRM

Figure 7: Visualize the distribution of item embeddings on the Amazon-Book dataset using DDRM, HICF, and HDRM. HDRM ensures that popular and unpopular items have representations with almost the same positions in the same space.

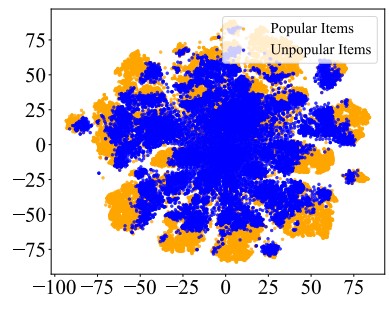 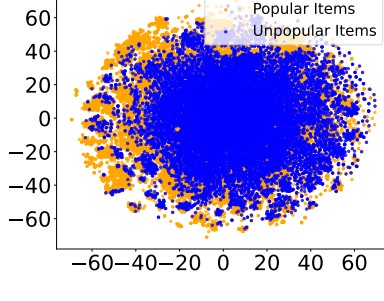 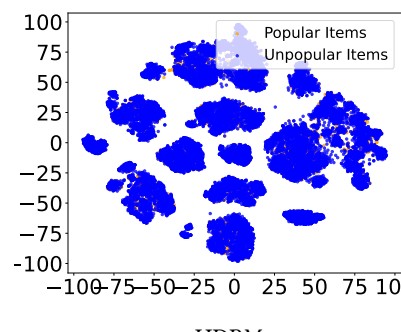

DDRM · HICF · HDRM

Figure 8: Visualize the distribution of item embeddings on the Yelp2020 dataset using DDRM, HICF, and HDRM. HDRM ensures that popular and unpopular items have representations with almost the same positions in the same space.

the representation space. In contrast, HDRM achieves a more uniform distribution of both types of embeddings within the same space. This observation suggests that HDRM effectively mitigates the tendency of recommender systems to over-recommend popular items at the expense of niche selections. Interestingly, HICF demonstrates a more pronounced differentiation between the two embedding categories. This characteristic can be attributed to the curvature of hyperbolic space, which allows for exponential growth of representational capacity within a finite area. Consequently, this property naturally amplifies item distinctions, particularly in terms of popularity.

