# OpenReview forum: "Hyperbolic Diffusion Recommender Model"
_ACM.org/TheWebConf/2025/Conference — WWW 2025 Poster_

### Official Review · Reviewer_2YKE · 2024-11-07

**Novelty:** 4
**Technical Quality:** 4

**Review:**

This paper focuses on the diffusion-based recommender models and proposes to address the anisotropic data structures in regular diffusion-based recommender systems with hyperbolic diffusion model. The proposed HDRM enhances the DDRM with an HGCN recommender backbone and a geometric constraint diffution process, which encourages the hyperbolic radial growth of diffused distribution. The model performance of HDRM is validated through comprehensive experiments.

Pros.
+ A pioneering work that explores the geometry of interaction data samples.
+ A novel method that introduces hyperbolic diffusion into recommendation tasks.
+ The proposed model is well presented with open-sourced implementation.

Cons.
- The overall framework of HDRM is more of an incremental adaptation of previous DDRM by integrating hyperbolic GCNs and hyperbolic diffusion process.
- The ablation study indicates the superior performance of HDRM can still be achieved even without either the hyperbolic encoder or the diffusion process, conflicting with the conclusion of DDRM that the recommender backend model is decisive for model performance.
- The training of HDRM requires simulating the reverse sampling process, which is time consuming. The execution time and training dynamics should be included to illustrate the stability and efficiency of HDRM in training.
- The huge scatter plots in this paper frequently slows down document browsing. It would be better to replace those with JPG or other compressed formats.

**Questions:**

1. Can you elucidate the detailed settings of ablation studies?  The ablation results seem quite unsual if the ablation of hyperbolic encoder means replacing the HGCN with mere node embeddings,
2. The calculation of $\mathcal{L}_{re}$ requires a sampling process starting from a uniformly sampled $t$. However the complexity in section 3.4.1 only involves a one-shot sampling from marginal distribution. Could you explain this?

**Reviewer Confidence:**

4: The reviewer is certain that the evaluation is correct and very familiar with the relevant literature

**Scope:**

3: The work is somewhat relevant to the Web and to the track, and is of narrow interest to a sub-community

---

### Official Review · Reviewer_eztn · 2024-11-28

**Novelty:** 6
**Technical Quality:** 5

**Review:**

**Strengths**:
- Novelty: The authors propose an HDRM, which is the first work to design a hyperbolic diffusion model for recommender systems. In my opinion, an interesting aspect is that, unlike existing Euclidean-based directional propagation methods, HDRM utilizes the natural geometric properties of hyperbolic space to achieve directional diffusion propagation by constraining the radial and angular components.
- Theoretical analysis: The paper provides a thorough analysis of the theories underlying hyperbolic spaces and diffusion models. Analyzing clustering and normal distributions in hyperbolic space, as well as examining constraints on directionality and angularity, presents an interesting avenue for exploration. The extensive theoretical proofs reflect the authors' in-depth exploration of a wide range of literature.
- Experiments: The experimental results are extensive and detailed, and the code is reproducible.

**Weakness**:
- To my knowledge, hyperbolic methods demonstrate robust performance even in low-dimensional embeddings. Did the authors explore HDRM's performance under low-dimensional embedding settings? Additionally, did they experiment with trainable curvature and provide comparative results for different curvature settings?
- The authors should provide a more detailed explanation of Fig. 3 to aid reader comprehension.
- Please verify whether the plus sign on line 568 and the dash on line 505 are appropriately formatted.

**Questions:**

Please see the above weakness

**Reviewer Confidence:**

3: The reviewer is confident but not certain that the evaluation is correct

**Scope:**

4: The work is relevant to the Web and to the track, and is of broad interest to the community

---

### Official Review · Reviewer_dsQa · 2024-11-28

**Novelty:** 6
**Technical Quality:** 5

**Review:**

**Summary:**

This paper introduces a novel Hyperbolic Diffusion Recommender Model (HDRM) to address anisotropic diffusion processes in recommendation systems. HDRM leverages the natural geometric properties of hyperbolic space, constraining radial and angular components to promote directional diffusion propagation. This ensures the preservation of the original topological structure in user-item interaction graphs. Extensive experiments on three datasets demonstrate the effectiveness of HDRM.

**Strengths:**
1. Novel application of hyperbolic geometry to diffusion models, addressing structural disparities in user-item graphs.
2. Robust performance under noisy data conditions.
3. Comprehensive evaluation on diverse datasets with extensive ablation studies.

**Weaknesses:**
1. Observational data are biased, and HDRM facilitates directional diffusion propagation may amplifying biases in existing recommender systems (e.g., popularity bias)?
2. There is a lack of experimental verification of whether hyperbolic geometry space and geometric constraints exacerbate the bias problem (e.g., popularity bias).
3. No algorithmic limitations were written.
4. HDRM facilitates directional diffusion propagation may amplifying fairness issues in existing recommender systems?
5. The high computational complexity of the hypergeometric embedding and diffusion process, especially on large-scale user-item interaction graphs, may lead to excessive training time and resource consumption, affecting the scalability of practical applications.

**Questions:**

Please see the weaknesses.

**Reviewer Confidence:**

2: The reviewer is willing to defend the evaluation, but it is likely that the reviewer did not understand parts of the paper

**Scope:**

4: The work is relevant to the Web and to the track, and is of broad interest to the community

---

### Official Review · Reviewer_LteD · 2024-11-29

**Novelty:** 4
**Technical Quality:** 5

**Review:**

This paper is technically solid, with a well-grounded approach and effective results, but its writing is confusing in several areas and requires significant improvement for better clarity and accessibility.

Strength

1. The paper effectively adapts the diffusion process to hyperbolic spaces, specifically designed to handle the inherent anisotropic nature of items in recommender systems.

2. The model design is rigorously grounded in hyperbolic geometry, with clear theoretical support

3. The experiments conducted on multiple benchmark datasets show the effectiveness of framework. Ablation studies further analyze the contributions of different components, strengthening the model's credibility.


Weakness

1. The writing needs improvement. There are many confusing typos:
- The quality of the framework figure is poor, lacking details of the model design. There are also some typos, e.g. "inverse process" instead of "reverse process; $\mathbf{Z}$ is not introduced in the paper but used in the plot.
-  Notation inconsistencies include the reuse of r in equations (9) and (11), and the loss balance parameter being defined as 𝛼 in the main text (Equation 20) but as 𝜆 in Appendix B.1.2.
- Some details of the diffusion model is not specified, such as the structure used for the denoising module $\mu_\theta$ and $\Sigma_\theta$, and the choise of noise schedule.
-Some parameter setting or serach range such as $r$, $\beta$ is not specified
- where is denoising weight factors $\gamma$ used?

2. Discussion about the complexity of the reverse process is missing in the complexity analysis.  Reverse process requires iterative inference and denoising, which is the main source of the complexity of the diffusion model.

3. There should also be sensitivity analysis about other essential parameters (e.g. the loss balance parameter, $\beta$)

**Questions:**

Please refer to above for questions.

**Reviewer Confidence:**

3: The reviewer is confident but not certain that the evaluation is correct

**Scope:**

4: The work is relevant to the Web and to the track, and is of broad interest to the community

---

### Official Review · Reviewer_YF9h · 2024-12-04

**Novelty:** 3
**Technical Quality:** 4

**Review:**

In this paper, the authors propose a diffusion generative models based on the hyperbolic space for recommender systems.

Pros:
- The source code is available.
- Extensive experiments are conducted to show the effectiveness of the proposed method.

Cons:
- The technical contributions are relatively weak.
- The motivation of this paper is not very clear. Why dose hyperbolic space matter in the diffusion recommender models if we use anisotropic Gaussian noises following [1]?
- Evaluation experiments about the time complexity should be included.

[1] A Directional Diffusion Graph Transformer for Recommendation.

**Questions:**

1. It seems that the hyperbolic encoder has the smallest impact on models compared with the other components in the ablation study. Can authers give more explanaiton?
2. The authors had better give more details to explain the beneifts of Theorem 2.
Why is the improvements of N@20 relative low on Yelp2020?

**Reviewer Confidence:**

4: The reviewer is certain that the evaluation is correct and very familiar with the relevant literature

**Scope:**

3: The work is somewhat relevant to the Web and to the track, and is of narrow interest to a sub-community